# Molecular estimation of neurodegeneration pseudotime in older brains

Sumit Mukherjee [1,8,10], Laura Heath[1,10], Christoph Preuss[2], Suman Jayadev[3], Gwenn A. Garden[3], Anna K. Greenwood[1], Solveig K. Sieberts [1], Philip L. De Jager [4,5], Nilüfer Ertekin-Taner[6,7], Gregory W. Carter [2], Lara M. Mangravite[1] & Benjamin A. Logsdon [1,9✉]

The temporal molecular changes that lead to disease onset and progression in Alzheimer's disease (AD) are still unknown. Here we develop a temporal model for these unobserved molecular changes with a manifold learning method applied to RNA-Seq data collected from human postmortem brain samples collected within the ROS/MAP and Mayo Clinic RNA-Seq studies. We define an ordering across samples based on their similarity in gene expression and use this ordering to estimate the molecular disease stage–or disease pseudotime-for each sample. Disease pseudotime is strongly concordant with the burden of tau (Braak score, $P = 1.0 \times 10^{-5}$), Aβ (CERAD score, $P = 1.8 \times 10^{-5}$), and cognitive diagnosis ($P = 3.5 \times 10^{-7}$) of late-onset (LO) AD. Early stage disease pseudotime samples are enriched for controls and show changes in basic cellular functions. Late stage disease pseudotime samples are enriched for late stage AD cases and show changes in neuroinflammation and amyloid pathologic processes. We also identify a set of late stage pseudotime samples that are controls and show changes in genes enriched for protein trafficking, splicing, regulation of apoptosis, and prevention of amyloid cleavage pathways. In summary, we present a method for ordering patients along a trajectory of LOAD disease progression from brain transcriptomic data.

[1] Sage Bionetworks, Seattle, WA, USA. [2] The Jackson Laboratory, Bar Harbor, ME, USA. [3] Department of Neurology, University of Washington, Seattle, WA, USA. [4] Center for Translational & Computational Neuroimmunology, Department of Neurology, Columbia University Irving Medical Center, New York City, NY, USA. [5] Taub Institute, Columbia University Irving Medical Center, New York City, NY, USA. [6] Department of Neurology, Mayo Clinic Florid, Jacksonville, FL, USA. [7] Department of Neuroscience, Mayo Clinic Florida, Jacksonville, FL, USA. [8] Present address: Microsoft, Redmond, WA, USA. [9] Present address: Cajal Neuroscience, Seattle, WA, USA. [10] These authors contributed equally: Sumit Mukherjee, Laura Heath. ✉email: ben@cajalneuro.com

Late-onset Alzheimer's disease (LOAD) is a devastating illness with no effective disease modifying therapy, owing to a 99.6% failure of clinical trials[1]. There is a growing consensus that the most effective treatments will intervene early in disease progression and halt disease pathophysiological processes prior to conversion to LOAD[2]. In addition, there is increasing recognition that LOAD may in fact be a spectrum of related diseases that have similar clinical and neuropathological manifestations[3,4]. Devising successful therapeutic strategies will likely require targeting potentially diverse early stage disease processes that occur prior to a high burden of neuropathology or cognitive impairment.

Current approaches to identify AD affected individuals include in vivo measures of the pathological hallmarks of disease—amyloid, tau, and neurodegeneration—via CSF biomarkers for amyloid and tau[5], positron emission tomography for amyloid and tau[6], and structural and functional MRI of neurodegeneration. Cognitive assessments are used to estimate disease burden[7], although measurable cognitive impairment generally indicates a sustained burden of neuropathology and advanced neurodegeneration. Based on biomarker studies of AD, by the time cognitive decline becomes detectable, neuropathological changes of AD have already occurred, first in Aß and subsequently in tau-related measures[8] and therefore cannot be used to select patients for early disease stage studies. Furthermore, while these measures of disease progression capture the overall increase in burden of pathology and cognitive decline, they do not necessarily identify the dysfunctional molecular mechanisms that lead to neuropathology and cognitive decline. There are likely many independent patient-specific molecular pathways present at an early stage in disease that then contribute to later stage disease progression[9,10]. This motivates the need to identify these early stage molecular mechanisms driving disease progression.

The Accelerating Medicines Partnership-Alzheimer's Disease (AMP-AD) consortia have generated genome-wide transcriptomics of postmortem brain tissue from patients across a broad range of Alzheimer's disease (AD) neuropathological progression—including individuals with various stages of AD neuropathology and those who lack AD neuropathology, but who may in fact harbor early stage disease molecular processes. We therefore sought to chart the molecular progression of the disease as reflected in the aggregate behavior of the brain transcriptome across these individuals. While standard approaches such as differential expression or co-expression analyses have proven informative[11–15], these analyses do not infer the relative stage of disease progression or identify distinct disease subtypes. Here we propose an approach to analyze population level RNA-seq data from postmortem brain tissue to learn a tree structured progression (Fig. 1) that represents distinct subtypes of disease and the relative progression of disease across patients. With this approach, we identify potentially generalizable trajectories of LOAD across heterogeneous patient populations at all stages of disease. Furthermore, we characterize molecular pathways that define disease stages—a potential source of biomarkers and therapeutic interventions for early stage disease processes along multiple different disease trajectories.

To learn the molecular disease staging and neuropathologic progression tree we use a manifold learning method[16]. Manifold learning refers to a group of algorithms that aim to recover the low-dimensional subspace underlying a high-dimensional data set. Previous authors use manifold learning to estimate disease progression from neuroimaging data[17] and to study lineage commitment of cells during differentiation from single cell RNA-seq (scRNA-seq)[18–21]. To our knowledge, manifold learning has not been used to estimate disease progression and/or disease stages from bulk RNA-seq data derived from postmortem brain tissue. Henceforth, we refer to manifold learning and lineage inference interchangeably in reference to the construction of a disease progression tree. We demonstrate that these tools can estimate the disease staging and progression tree (Fig. 2) from bulk RNA-Seq data collected from postmortem brain tissues in a case/control cohort. Moreover, these trees show clear LOAD staging, enable the study of cell-type-specific effects of LOAD, and allow the identification of genetic factors driving disease progression.

## Results

**Manifold learning distinguishes pathologically defined LOAD from control.** We first quantify the bulk RNA-Seq data from the ROS/MAP and Mayo Clinic cohorts into gene counts and remove any batch effects introduced due to sequencing runs using standard count normalization (see "Methods"). The data from the ROS/MAP cohort are sampled from the dorsolateral prefrontal cortex (DLPFC), and the data from the Mayo Clinic cohort are sampled from the temporal cortex (TCX). Patient's clinical characteristics are reported in Supplementary Table 1 and described in "Methods." The full pipeline we used for RNA-Seq data generation and quality control was recently reported[22]. The entire transcriptome comprises many genes, which do not have measurable expression or vary across case/control samples, which we remove in order to reduce the noise in manifold learning[19]. To do this, we first perform differential expression analysis between case/control samples separately for each study and retain genes that reach an FDR of 0.10. To test if this biased the disease lineage inference, we also perform manifold learning using only genes with high variance across samples, and we see a strong concordance with disease lineages inferred with differentially expressed genes (Supplementary Fig. 1). Changing the significance threshold for the differential expression analysis to FDR < 0.01 did not materially change these results (Supplementary Fig. 2). We infer the disease lineage for each brain region on this subset of retained genes (Fig. 2a, b). Adjusting for postmortem interval (PMI) (Supplementary Figs. 3A and 4A), ten principal components from a principal component analysis (PCA) of genotype data to account for ancestry effects (Supplementary Figs. 3B and 4B), RNA integrity number (RIN) (Supplementary Figs. 3C and 4C), or all of these variables (Supplementary Figs. 3D and 4D) did not materially change the overarching ordering of patients for either the TCX or DLPFC regions. Furthermore, to assess the general robustness of the results, we apply leave one out cross validation to infer disease pseudotime for both DLFPC and TCX brain regions and find strong correlations between lineages inferred with each sample removed, and the lineage for the entire sample set (Supplementary Fig. 5).

We visualize the clinical diagnosis of the samples on the inferred disease staging tree to verify that there is indeed separation of AD patients across the tree. To determine if inferred tree structure is an accurate model of disease progression, we introduce the notion of disease pseudotime, which is the geodesic distance along the tree from an inferred initial point to the point of interest as a quantitative linear measure of LOAD stage. We scale this estimated disease pseudotime to lie in the range [0,1] to make the effects comparable between the two studies (and brain regions). We show that for LOAD cases compared to controls there is a significant association ($P = 0.02$ in Mayo and $P = 2.0 \times 10^{-6}$ in ROS/MAP, logistic regression) between the estimated pseudotime and AD case/control status (Fig. 2c). These effects are not abrogated by adjusting for RIN, PMI, or ancestry in either tissue (Supplementary Fig. 6A–D and Supplementary Table 2). To assess whether the association between inferred disease pseudotime is a phenomena in only the Mayo RNA-seq and

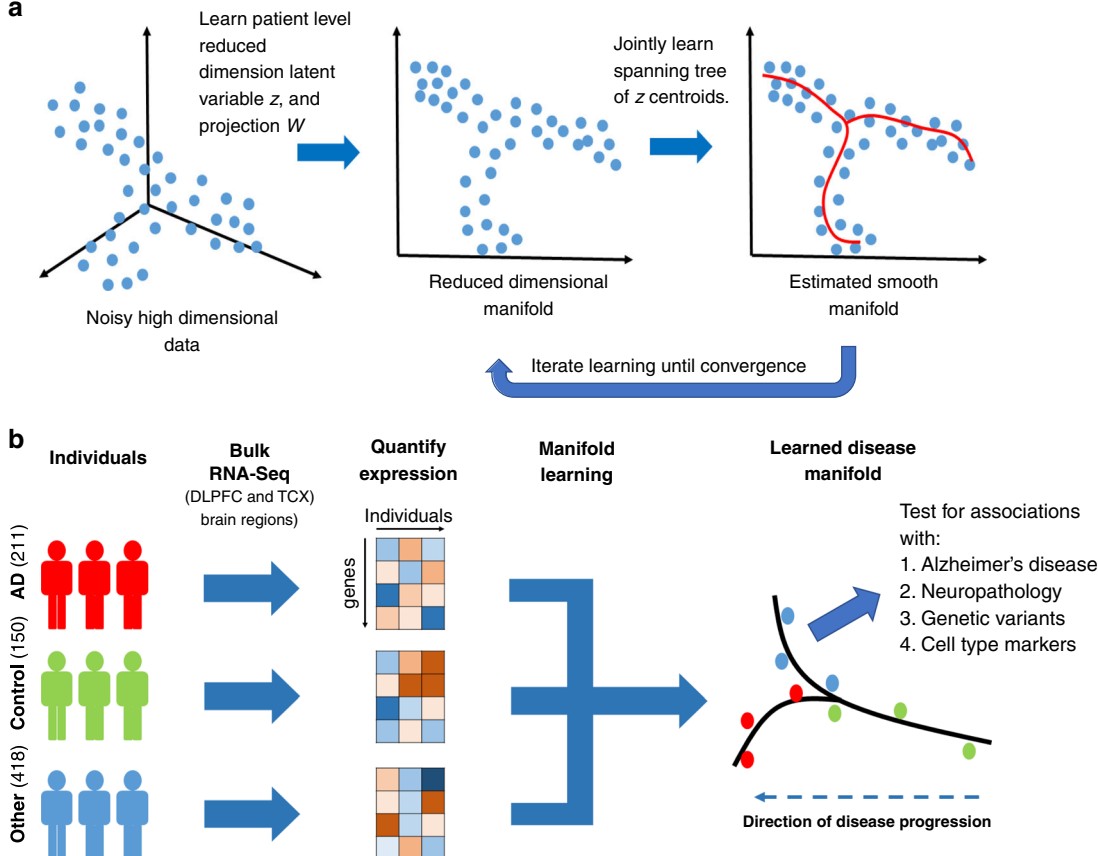

**Fig. 1 Overview of manifold learning for unraveling staging in Alzheimer's disease. a** Illustration of steps in manifold learning using reverse graph embedding DDRTree method. **b** Illustration of lineage inference process for LOAD. RNA-seq samples with different disease diagnoses were pooled, batch normalized, and a smooth manifold was learned for each brain region across individuals (each point is an individual). Total sample numbers are indicated across Mayo RNA-seq TCX and ROS/MAP DLPFC for the different diagnoses in parentheses.

ROS/MAP RNA-seq data, we also apply the lineage inference approach to expression array data from the Mayo eQTL study[23] (see "Methods"). These samples are derived from a completely independent set of donors than the Mayo RNA-seq study[24]. Similarly, we restrict to only female samples, and test for an association between inferred disease pseudotime and disease status (Supplementary Fig. 7A, B). We see a significant association between disease pseudotime and neuropathological AD diagnosis ($P = 2.2 \times 10^{-8}$).

Furthermore, we observe strong evidence of sex heterogeneity when performing the manifold learning approach and find that the manifolds inferred for female only samples show stronger association with pseudotime than for male samples. This matches previous observations concerning disease-specific sex heterogeneity[22]. As such, we do not see as statistically significant of an association between pseudotime and disease diagnosis in male samples ($P = 0.040$ in Mayo and $P = 0.11$ in ROS/MAP, logistic regression, Supplementary Fig. 8A–D). Similarly, the association between pseudotime and amyloid, tau, and cognitive diagnosis is attenuated in male samples in ROS/MAP (Supplementary Fig. 9A–D). For the combined samples, we see moderate evidence of disease association with pseudotime ($P = 0.003$ in Mayo and $P = 0.003$ in ROS/MAP, logistic regression, Supplementary Fig. 10A–D). The association with neuropathological measures of disease is more robust in the combined sample (Supplementary Fig. 11A–F), but not as strong as in females only, hence we restrict to female only analyses for all subsequent reported results.

We test whether genes in loci that have been implicated in genome-wide association studies of LOAD are associated with inferred disease pseudotime. We use the prioritized LOAD GWAS genes[25], Supplementary Table 3, and compute the correlation between their expression and inferred pseudotime (Fig. 2d). When compared to the background of all genes, we see that there is a significant increase in positive correlation with disease pseudotime for implicated LOAD GWAS genes ($P$ value: $7.3 \times 10^{-5}$ in Mayo and $5.6 \times 10^{-3}$ in ROS/MAP). This effect is robust to adjustments for PMI, RIN, or ancestry (Supplementary Fig. 12A–D). Furthermore, this does not appear to be driven by a small subset of outlier genes, but by the majority of the distribution of LOAD GWAS genes. The fact that AD GWAS loci genes have expression associations with pseudotime likely implies that the AD risk variants at these are also eQTL as previously shown[26–29] and/or are members of co-expression networks that are differentially expressed in AD[13,30].

To further explore the relationship between inferred disease stage and LOAD, we test for its association with neuropathological and clinical measures of LOAD severity, namely: (1) Braak score, (2) CERAD score, and (3) cognitive diagnosis. The ROS/MAP study has numeric scores for these categories available as covariates for each sample. Braak is a semiquantitative measure that increases with tau pathology[31] and CERAD is a semiquantitative measure of density of neuritic plaques[32]. We overlay these scores on the inferred manifold for the DLPFC brain region (Fig. 3a). We observe a progressive increase in tau, amyloid, and cognitive burden as we traverse the inferred disease manifold (Fig. 3a). This is further quantified by characterizing the relationship between branches of the inferred manifold and Braak, CERAD, and cognitive diagnosis (Fig. 3b). We observe

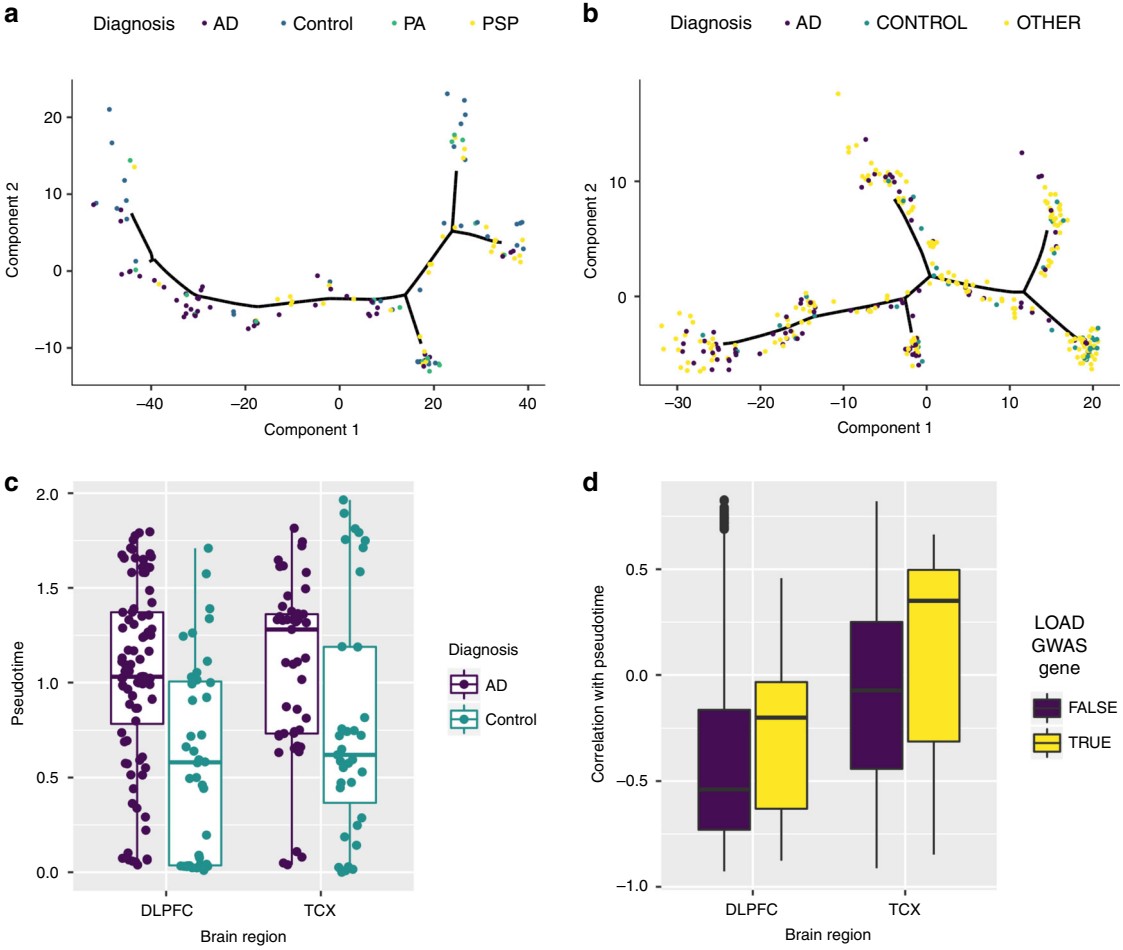

**Fig. 2 Manifold learning accurately infers disease states and stages from RNA-seq samples. a** Estimated disease progression trees from temporal cortex (TCX) and **b** dorsolateral prefrontal cortex (DLPFC) brain regions showing localization of identified LOAD samples on particular branches. **c** Distribution of pseudotime for AD cases and controls for both DLPFC and TCX for 218 independent samples from two independent studies. **d** Distribution of expression correlation with pseudotime for both LOAD GWAS genes and non-LOAD GWAS genes for 17446 genes from two independent studies. Box plots have lower and upper hinges at the 25th and 75th percentiles and whiskers extending to at most 1.5xIQR (interquartile range).

significant associations between pseudotime and Braak score ($P = 1.0 \times 10^{-5}$), CERAD score ($P = 1.8 \times 10^{-5}$), and cognitive diagnosis ($P = 3.5 \times 10^{-7}$). In ROS/MAP, adjustment for Braak score when fitting the discriminative dimensionality reduction tree (DDRTree) method attenuates the association between pseudotime and disease states (P value: 0.214, Supplementary Fig. 13A, B), though there is still evidence of association with cognitive diagnosis (P value: 0.03, Supplementary Fig. 13C, D). In the Mayo RNA-seq study, we have Braak score and Thal Amyloid scores for only a subset of samples but observe a similar pattern as in ROS/MAP (Supplementary Fig. 14A, B) for the samples that we do have data. There is a significant association between Braak score and pseudotime (P value: $5 \times 10^{-5}$) as well as Thal amyloid (P value: $1.7 \times 10^{-5}$) within this subset with available neuropathology data.

**Comparison to other unsupervised learning approaches**. We compare the manifold learning approach to other unsupervised learning approaches including PCA, t-distributed stochastic neighbor embedding (tSNE)[33], and Uniform Manifold Approximation and Projection (UMAP)[34]. Correlations between the first two dimensions of each of these approaches and the DDRTree learned pseudotimes are shown in Supplementary Figs. 15A–F and 16A–F for DLPFC and TCX brain tissues, respectively. We see the strongest correlations between PCA1 and UMAP2 and

pseudotime in both data sets, increasing our confidence that the overarching ordering of patients along a disease pseudotime is a robust characteristic of the disease progression as reflected in the gene expression changes as a function of disease, and not dependent on the underlying manifold learning approach. This is further supported by inspecting the association between these approaches and Braak, CERAD, and cognitive diagnosis (cogdx) scores for the DLPFC tissue (Supplementary Table 4). Furthermore, the manifold learning approaches (DDRTree and UMAP) have much stronger associations with Braak, CERAD, and cogdx scores than either PCA or tSNE. In fact, UMAP has been proposed for lineage inference[35], and when we apply UMAP with lineage inference using Monocle3, we observe similar results (Supplementary Figs. 17 and 18) as with DDRTree and Monocle 2 (Figs. 2 and 3), though the inferred pseudotimes from Monocle3 are not quite as significant as the association with UMAP2 or from Monocle 2 with Braak, CERAD, and cogdx (Supplementary Table 4).

**Inferred staging recapitulates known biology of AD**. To demonstrate that the inferred disease pseudotime recapitulates known biology of LOAD, we test for association between inferred disease stage and both the cellular response to disease and the genetics of the disease. A prominent hypothesis in AD is that the effects of the disease vary across different brain cell types,

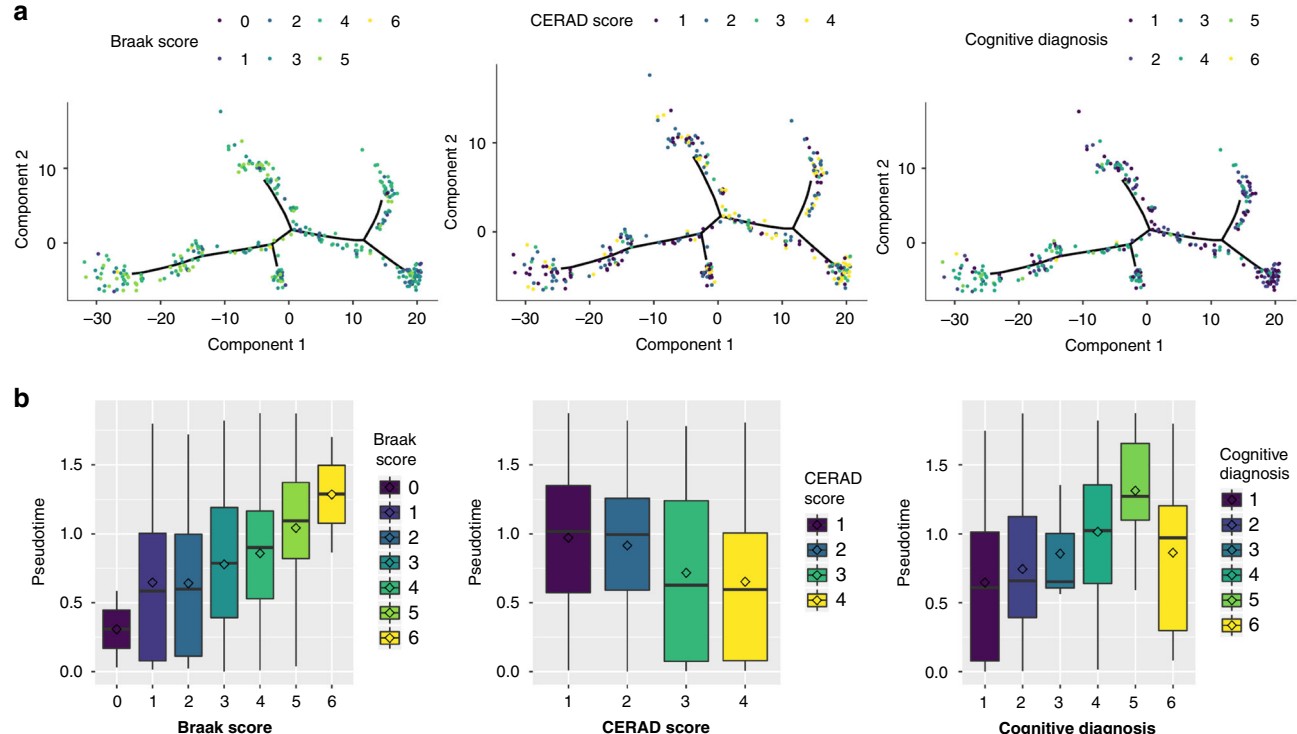

**Fig. 3 Manifold learning replicates existing measures of staging in LOAD in DLPFC samples. a** Samples colored by three different external measures of LOAD staging, namely: Braak score (tau pathology), CERAD score (amyloid pathology), and cognitive diagnosis (clinical measure of disease severity). Black lines denote inferred lineages. **b** Distribution of samples by inferred stage for different distinct stages in each of the three methods of measuring LOAD severity for 338 independent samples from one study. Inferred disease stages generally corresponded with all methods, and cognitive diagnosis demonstrated the strongest alignment. Box plots have lower and upper hinges at the 25th and 75th percentiles and whiskers extending to at most 1.5xIQR (interquartile range).

specifically neurons and glial subtypes. Current understanding of the cell biology of the disease implicates progressive neuronal loss and increase in gliosis[36]. To test if the inferred pseudotime aligns with existing cell-type-specific hypotheses regarding AD, we first selected from the genes used in lineage construction the marker genes for four key cell types: neurons, astrocytes, microglia, and oligodendrocytes based on a previously published brain cell atlas[37] (Supplementary Table 5). We then calculate the normalized mean expression for the marker genes of each cell type and fitted a linear model to the mean expression with disease pseudotime as the dependent variable. We find that, in both studies, the cell-specific marker gene levels show a statistically significant linear dependence on pseudotime (Supplementary Table 6). Fitted effects recapitulate known neuropathologic changes which occur in AD, namely: (1) a reduction in the neuronal populations as AD progresses, and (2) an increase in expression associated with activation of microglia, astrocytes, and oligodendrocytes as AD progresses (Fig. 4).

Next, we test for association between assigned lineage state in ROS/MAP (DLPFC) and Mayo (TCX) and APOE e4 status (Supplementary Fig. 19). For reference, the inferred trees for TCX and DLPFC each resolve into six branches (Figs. 5a and S20). Carriers of the APOE e4 allele are significantly enriched on the State 4 branch in TCX (*P* value = 0.027, unadjusted), and suggestively enriched on the State 5 branch (*P* value = 0.06, unadjusted), compared to the State 1 branch (logistic regression). Similarly, in the Mayo eGWAS study, when we perform an ordinal logistic regression of APOE e4 dosage and disease pseudotime we see a significant positive association as a function of pseudotime (*P* value = $6.9 \times 10^{-4}$, Supplementary Fig. 7C).

**Genetic factors associated with inferred disease staging.** Lineage inference of LOAD transcriptomes provides a quantitative measure of disease progression for genetic associate testing, and the significantly greater correlation between pseudotime and gene expression for known LOAD risk genes (Fig. 2d) suggests that the observed differences in disease trajectories are influenced by genetic factors. To test this hypothesis, we perform single variant analysis using whole-genome sequencing data for 305 patients from the ROS/MAP and 131 patients from the Mayo cohort. Despite the limited sample size, resulting in lack of statistical power to discover genome-wide significant associations, multiple variants reach a genome-wide suggestive threshold of $P < 1 \times 10^{-5}$ (Supplementary Table 7). We do not see evidence of population stratification in the analysis (Supplementary Figs. 21–22). Notably, the most significant association with pseudotime for the ROS/MAP cohort is observed at the *PTPRD* locus (rs7870388, $P = 1.31 \times 10^{-6}$) (Supplementary Fig. 23 and Supplementary Table 7). The *PTRPD* locus is associated with the susceptibility to neurofibrillary tangle independent of amyloid deposition in the ROS/MAP cohort[38]. For the Mayo Clinic cohort, known LOAD variants in the *APOE* (rs6857, $P = 9.18 \times 10^{-6}$) and *BIN1* (rs62158731, $P = 4.68 \times 10^{-5}$) loci overlap with variants associated with inferred disease stage (Supplementary Fig. 23 and Supplementary Table 7)[39]. When comparing our association results for inferred disease stage with summary statistics from a large-scale case-control approach, we identify multiple variants that have been previously associated with LOAD in the International Genetics of Alzheimer's Project (IGAP) cohort (Supplementary Table 8). Furthermore, we identify several genes associated with inferred disease stage (*ADAMTS14*, *IL7*, and *MAN2B1*) linked to immune

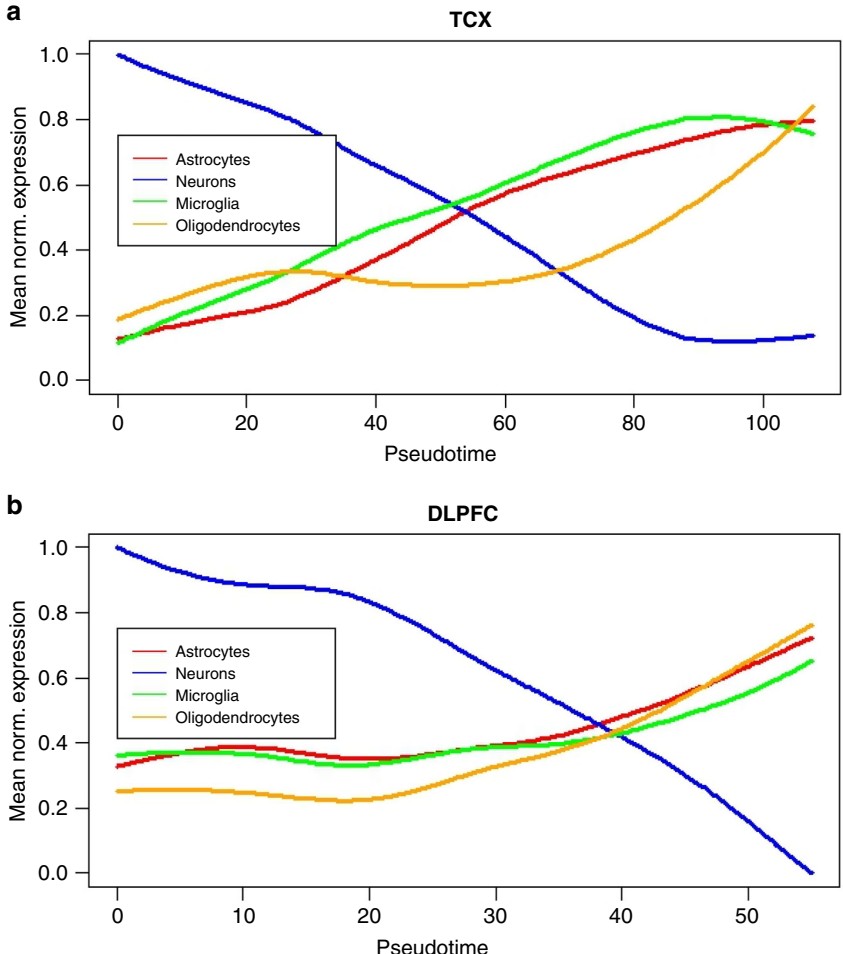

**Fig. 4 Cell-type gene expression signatures as a function of disease pseudotime. a** Mean expression of cell markers for astrocytes, neurons, microglia, and oligodendrocytes as a function of pseudotime for TCX brain region, **b** mean expression of cell markers for astrocytes, neurons, microglia, oligodendrocytes as a function of pseudotime for DLPFC (**b**) brain region.

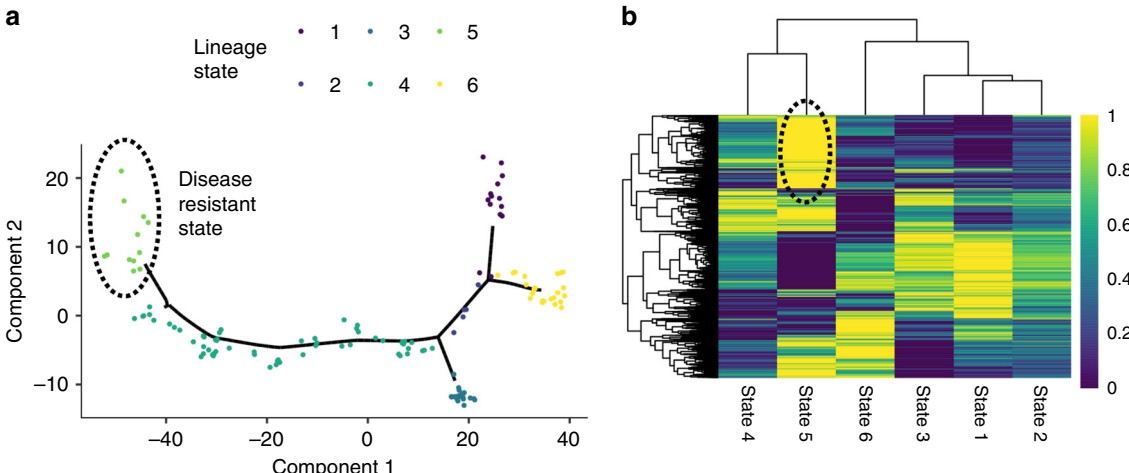

**Fig. 5 Disease resistant state. a** The inferred manifold from the TCX region with samples colored by their inferred disease subtype/state is shown in the left panel. State 5 (dots, circled) lies at the late end of the disease trajectory, indicating a strong disease-like transcriptomic phenotype, yet most samples in the group did not have pathologically diagnosed AD (Fig. 2a). We hypothesize this group represents a disease resistant state to the disease. **b** Biclustering results of average expression from each disease state, with increased expression of a gene cluster (Cluster 4) unique to State 5 is shown in the right panel.

and lysosomal storage function (Supplementary Fig. 23 and Supplementary Table 7). *IL7* has been proposed as an inflammatory biomarker for LOAD that correlates with disease outcome and severity[40]. *ADAMTS14* is part of a locus that has been previously linked with Alzheimer susceptibility and plays an important role in the regulation of immune function via TGF-beta signaling.

**Table 1 Pathway enrichments for branch-specific differentially expressed genes in TCX.**

| Direction | Branch | Representative enriched Gene Ontology terms |
|---|---|---|
| Down | 2 | prespliceosome (GO:0071010), mitochondrial electron transport cytochrome c to oxygen (GO: 0006123) |
| Down | 3 | negative regulation of microtubule, polymerization or depolymerization (GO:0031111) |
| Down | 4 | mitochondrial electron transport, NADH to ubiquinone (GO: 0006120), spliceosomal tri-snRNP complex (GO:0097526), negative regulation of microtubule depolymerization (GO:0007026) |
| Down | 5 | axon (GO:0030424), protein kinase C activity (GO:0004697) |
| Down | 6 | gamma-tubulin large complex (GO:0000931), U1 snRNP (GO:0005685), mitochondrial respiratory chain complex IV (GO:0005751), response to cadmium ion (GO:0046686) |
| Up | 3 | fatty acid elongase activity (GO:0009922), ubiquitin protein ligase activity (GO:0061630) |
| Up | 4 | transforming growth factor beta-activated receptor activity (GO:0005024), hippo signaling (GO:0035329), regulation of extrinsic apoptotic signaling pathway via death domain receptors (GO: 1902041), regulation of DNA repair (GO: 0006282) |
| Up | 5 | regulation of apoptotic process (GO:0042981), leptin mediated signaling pathway (GO:0033210), negative regulation of hippo signaling (GO:0035331), small GTPase binding (GO:0031267) |
| Up | 6 | extracellular ligand-gated ion channel activity (GO:0005230), integral component of mitochondrial inner membrane (GO:0031305) |

Differential expressed genes are identified with a two-sided Tukey's honest significant difference test (FDR < 0.05), with Branch 1 as the reference, and pathway enrichments that are significant from a one-sided Fisher's exact test are shown (FDR < 0.05).

**Table 2 Pathway enrichments for branch-specific differentially expressed genes in DLPFC.**

| Direction | Branch | Representative enriched Gene Ontology terms |
|---|---|---|
| Down | 2 | DNA repair (GO:0006281), intracellular protein transport (GO:0006886) |
| Down | 3 | mismatch repair complex binding (GO:0032404) |
| Down | 5 | mitochondrial respiratory chain complex assembly (GO: 0033108) |
| Up | 2 | racemase and epimerase activity (GO: 0016857) |
| Up | 3 | racemase and epimerase activity (GO: 0016857) |
| Up | 4 | vesicle mediated transport (GO: 0016192) |
| Up | 5 | NuRD complex (GO: 0016581) |
| Up | 6 | microtubule motor activity (GO:0003777), AP-2 adapter complex binding (GO:0035612) |

Differential expressed genes are identified with a two-sided Tukey's honest significant difference test (FDR < 0.05), with Branch 1 as the reference, and pathway enrichments that are significant from a one-sided Fisher's exact test are shown (FDR < 0.05).

**New disease insights identified from inferred disease lineages**. Another important direction of study in the field of Alzheimer's is the identification of disease subtypes, which has so far predominantly been done using imaging data[41]. The branches of the inferred disease trees provide a transcriptomic-based approach to identify disease subtypes. In both brain regions and in two separate cohorts, there were two distinct early lineage branches corresponding to predominantly control samples, which we interpret as different initial paths toward the disease. Similarly, both brain regions feature several distinct branches with predominantly LOAD samples (Fig. 2a, b).

**Branch-specific differential expression patterns**. To study the genes and pathways specific to each branch, we perform a branch-specific differential expression analysis with an ANOVA model using the branches with the highest proportion of controls as the reference branch for DLPFC (Supplementary Table 9) and TCX (Supplementary Table 10). We see many genes are differentially expressed between the control branch and branches that are enriched in the affected individuals (Supplementary Table 11). We test for overlap between the differentially expressed gene sets between the two studies (Supplementary Fig. 24), and find significant overlaps in branches enriched for late stage disease cases, especially between upregulated genes in State 6 of DLPFC and upregulated genes in State 5 of TCX (P value: $4.1 \times 10^{-108}$, OR: 4.5, Fisher's exact test), as well as genes that are upregulated in State 6 of DLPFC and in State 4 of TCX (P value: $1.8 \times 10^{-14}$, OR: 1.9, Fisher's exact test), and more modestly for genes that are

down regulated in State 6 of DLPFC and down regulated in State 3 of TCX (P value: $1.1 \times 10^{-6}$, OR: 1.6, Fisher's exact test). Next, we performed an enrichment analysis on each of these differentially expressed gene sets with the enrichR[42] package for Gene Ontology[43] annotations (see "Methods"). The results of this enrichment analysis for DLPFC and TCX tissues are shown in Supplementary Tables 12 and 13. Only gene sets with significant enrichment are shown (FDR adjusted P value < 0.05). Overall, we see a pattern of loss of expression of basic cell biology mechanisms in early stage branches including RNA splicing, mitochondria function, protein transport, and DNA repair. Late-stage branches were characterized by increased immune response (e.g., TGFb/WNT signaling) and apoptotic activity (Tables 1 and 2).

While studying the different branches in the two brain regions, we observe a branch (Branch 5) that corresponds to a group of predominantly neuropathological control samples from the Mayo RNA-seq cohort that were in close proximity to a branch with predominantly LOAD samples (Branch 4) on the inferred disease lineage (Fig. 5a). However, most of the samples on Branch 5 are neuropathological controls as defined by the Mayo diagnostic criteria. We bi-cluster the mean expression of genes in each branch and the branches themselves (see "Methods"). This clustering analysis (Fig. 5b) shows that the closest branch to this potentially disease resistant branch contains the highest proportion of AD samples. While the stage proximity implies some transcriptomic similarity between these controls and nearby cases, we also see a secondary cluster of genes with increased expression in the resistant state while having reduced expression in all other states. We perform an enrichment analysis on this set of genes and find significant GO

terms corresponding to protein transport (GO:0015031), regulation of mRNA splicing, via spliceosome (GO:0048024), negative regulation of apoptotic process (GO:0043066), and regulation of amyloid-beta clearance (GO:1900221) (Cluster 4, Supplementary Table 14). It is possible that these potentially disease resistant individuals have compensatory mechanisms, which suppress the hallmarks of disease despite sharing gene expression patterns with pathologically affected individuals.

To replicate this observation, we perform a differential expression analysis on individuals in the Mayo eGWAS study where we consider individuals that are in the top quintile of pseudotime but are classified as neuropathological controls as resistant individuals (Supplementary Fig. 7D, see "Methods"). To test if these individuals also have a similar resistant molecular endophenotype, we compare the overlap between various differential expressed gene sets derived from these resistant individuals and the gene sets identified in the biclustering of the Mayo RNA-seq data (Supplementary Fig. 25). We observe that there is a highly statistically significant overlap between genes that are upregulated in these Mayo eGWAS resistant individuals ($P$ value: $2.6 \times 10^{-51}$, OR: 2.9, Fisher's exact test), and the Cluster 4 genes that are upregulated in the Branch 5 Mayo RNA-seq samples (Supplementary Fig. 25).

## Discussion

Here we proposed an approach to infer the AD severity and disease subtypes in an unsupervised manner from postmortem bulk RNA-seq data that gets directly at the challenge of identifying the temporal progression of disease in the disease resistant tissue. Our strategy utilized a manifold learning approach to infer a disease progression tree from cross-sectionally collected patient samples from two different brain regions. The underlying assumption of our approach is that the inferred disease progression from cross-sectional samples serves as a proxy for the unobserved progression of the disease across subtypes of LOAD. We validated this hypothesis through comparisons with neuropathological measures of disease stage severity and against known cell-type-specific effects caused by the disease. While one could argue that the method is merely classifying patients as either disease cases or controls based on expression signatures of the hallmarks of disease, we see at least three advantages of this approach beyond that interpretation. First the application of this method appears to produce a more quantitative measure of disease state than strictly neuropathological assessments—as born out through the identification of distinct genetic loci that replicate based on IGAP summary statistics. This suggests that it may be adding information related to other aspects of disease such as the effect of neuroinflammation or neuronal injury. In addition, we see evidence of neuropathological controls that are disease resistant given their molecular state in two independent studies —which would not be detectable with standard neuropathological or clinical assessments—and could provide important molecular clues to mechanisms of disease resistance. Finally, there is the potential that specific pathways associated with early stage disease processes can be characterized, which is desperately needed for hypothesis generation in the field. Furthermore, this approach provides clues to better understanding the molecular heterogeneity of disease by identifying specific pathways that are dysregulated in subsets of patients at different disease stages. This opens up the possibility of better patient stratification and precision medicine.

We observed that different biological processes vary as a function of inferred disease stage, and that early stage disease processes—include RNA splicing, mitochondrial function, and protein transport—implicate multiple basic cell biology

mechanisms as potential early stage disease processes for further study in relevant model systems. Additionally, the manifold learning method identified six subtypes of LOAD from RNA-seq (i.e., branches), suggesting the LOAD populations should be stratified by better biomarkers with tailored treatment strategies. To identify and test these stratifications future studies could focus on longitudinal cohorts of patients with rich molecular and imaging data to identify biomarkers that can accurately and precisely stratify patients into the underlying molecular subtypes at different relative stages of disease. Furthermore, we observe a disease resistant subtype of patients. This disease resistance should be tested in disease model systems, to identify if neuro-pathological readouts can be modified by altering the function of the pathways identified in our analysis (e.g., APP processing, RNA splicing, apoptosis, and protein trafficking). While this preliminary observation needs to be validated in another cohort, it has the potential to be a source of hypotheses for therapeutic development. Specifically, for constructing better combination therapy hypotheses that may confer neuroprotection, even in patients that are mildly affected by disease.

LOAD is a complex and heterogeneous disease encompassing a broad spectrum of clinical symptoms. Disease progression can vary widely between patients leading to different rates of cognitive decline. Several lines of evidence suggest that these differences in progression are modified by multiple genetic factors affecting the transition from one pathological state to another[44,45]. However, it has remained difficult to assess the role of genetic variants affecting disease trajectories by case-control approaches alone. Here, we showed that our expression trait pseudotime might be used as a molecular phenotype to identify AD loci associated with different disease progression states across AD patients. Despite a limited sample size, we identified previously associated AD candidate loci in the Religious Orders Study and the Memory and Aging Project (ROSMAP) (*PTPRD*) and Mayo (*BIN1* and *APOE*) cohorts with suggestive significance ($P < 1 \times 10^{-5}$). Variants in *PTPRPD* have been associated with the susceptibility to neurofibrillary tangles, independent of amyloid burden. This is in line with the results from the differential gene expression analysis of pseudotime branches showing an enrichment of molecular pathways implicated in TAU pathology. Furthermore, our analysis revealed several loci linked to immune function (*ADAMTS14* and *IL7*) and neurotransmitter signaling (*CHRM2* and *CHRM3*) processes associated with disease pseudotime (Supplementary Table 4). Future studies will be needed to replicate these findings in independent cohorts of LOAD and validate the role of candidate genes in LOAD-related disease progression by first identifying peripheral biomarkers that correspond to this molecular definition of disease stage, and then testing for GWAS association with that disease stage. Subsequent results can improve functional interpretation by linking candidate genes with ordered pathological processes.

## Methods

**ROS/MAP and Mayo RNA-seq study population characteristics**. Detailed descriptions of cohort and patient characteristics included in this study can be found in previously published work[24,46]. Patient characteristics included in this study are summarized in Supplementary Table 1, stratified by sex. In brief: for Mayo samples, AD diagnosis was performed according to NINCDS-ADRDA criteria (probable or possible AD); control individuals had Braak NFT stage ≤ 3, CERAD score < 2.5, and lacked other pathologic diagnoses; Path.Aging are individuals who lacked any pathologic diagnoses and had Braak NFT ≤ 3 and CERAD score ≥ 2. Progressive supranuclear palsy (PSP) individuals were diagnosed neuropathologically by a single neuropathologist (for further details, see[24]). For ROSMAP samples, AD diagnosis was according to NIA Reagan criteria, which combines neuropathology and clinical data; control individuals had no signs of cognitive impairment; and other individuals had MCI, mixed pathology, or other form of dementia. Age at death was collected for all patients in the ROSMAP study, though age at first AD diagnosis had a high degree of missingness and thus was not used as a variable in follow-up analyses (see Supplementary Table 1). Braak stage indicates the measure of severity of NFT pathology. Stages I and II indicate NFTs

confined mainly to the entorhinal region of the brain, stages III and IV indicate involvement of limbic regions, and stages V and VI indicate moderate to severe neocortical involvement. CERAD score is a semiquantitative measure of neuritic plaques. 1 = definite AD; 2 = probable AD; 3 = possible AD; 4 = no AD. Cognitive diagnostic category (as determined by neurologist): (1) NCI; (2) MCI and no other cause of CI; (3) MCI and another cause of CI; (4) AD and no other cause of CI; (5) AD and another cause of CI; (6) other primary cause of dementia[47]. Thal amyloid stages: phases of amyloid deposition. (0) no amyloid; (1) isocortical phase; (2) limbic phase; (3) basal ganglia phase; (4) basal forebrain and midbrain phase; (5) pons/medulla oblongata and cerebellum phase. The list of differentially expressed genes used to create the Monocle objects was based on results which included the whole data set.

**RNA sequencing**. The details of the sample collections, postmortem sample characteristics, the tissue and RNA preparations, the library preparations and sequencing technology and parameters, and sample quality control filters are provided in previously published work[24,46]. For the bioinformatic pipeline to produce gene-level counts, we applied a standard pipeline[22] where sequencing reads were aligned to the GENCODE24 (GRCh38) reference genome with STAR[48], and gene counts generated using the HTSeq algorithm[49]. Genes that had more than one counts per million total reads in at least 50% of samples in each tissue and diagnosis category were used for further analysis.

**Differential expression analysis on Mayo and ROS/MAP cohorts**. For gene filtering, we used false discovery rate of 0.05 from the previously published differential expression analysis of Mayo and ROS/MAP RNA-seq data[22]. Briefly, case-control status was harmonized across the Mayo and ROS/MAP cohorts, where controls were defined as individuals with a low burden of amyloid and tau based on CERAD and Braak scores, and cases with a high burden. Furthermore in ROS/MAP, clinical diagnosis was also used with controls having to have no cognitive impairment, and cases have probably AD[22]. Differential expression analysis was run on suitably normalized data—using conditional quantile normalization to account for variation in gene length and GC content, removing sample outliers, covariate identification adjustment, with sampling abundance confidence estimated using a weighted linear model with the voom-limma package[22,50,51]. A fixed/mixed effect linear model is used to fit the differential expression model on the normalized data[22].

**Manifold learning for LOAD**. Manifold learning refers to a group of machine learning algorithms that recover a low-dimensional subspace underlying a high-dimensional data set. Manifold learning approaches are typically used in data sets or applications where data samples lie on an underlying low-dimensional latent space (e.g., a tree, a line, and a curved plane). The low-dimensional space is learned via a projection from the high-dimensional space of the observed data (e.g., RNA-seq profiles across hundreds of patient samples) down to a low-dimensional space with suitable regularization constraints to enforce smoothness and the structural constraints of the low-dimensional space (Fig. 1a). Due to the necessary assumption of an underlying latent subspace, manifold learning is commonly used in applications where it is known that the observed data is obtained from a progression of some kind; e.g., (1) to infer the temporal ordering of a sequence of images, or (2) to infer the approximate lineage of cells in a differentiation trajectory using scRNA-seq data (Fig. 1b, c).

Here, we repurpose methods originally developed for learning cell lineage using scRNA-Seq data, to infer the staging of AD using bulk RNA-Seq data from postmortem brain samples with known AD diagnosis status. Since bulk RNA-Seq has many of the same sampling and distributional properties as scRNA-Seq, we observe that scRNA-Seq methods are applicable with no additional modifications. As such, we use the DDRTree manifold learning approach available in the Monocle 2 R package[19]. However, we also show that the estimated staging of disease is quite similar across some of the other common methods used for scRNA-Seq lineage estimation (Supplementary Fig. 26–28) including Monocle1[18] and diffusion pseudotime[52].

The RNA-Seq data used in this study were generated from postmortem brain homogenate samples, and obtained from two separate studies that are a part of the Accelerating Medicines Partnership in Alzheimer's Disease (AMP-AD) consortium, namely: (1) the ROSMAP[53,54], and (2) the Mayo RNA-seq study[24]. For this paper, we focused our analysis on the TCX and DLPFC tissue samples. Within the Mayo RNA-seq study, the TCX samples are derived from individuals neuropathologically defined as either aged controls, LOAD cases, PSP cases, or pathological aging cases[24]. The ROSMAP study is a prospective longitudinal cohort of an aging population, and has samples from participants with clinical and neuropathological diagnoses of LOAD[46], aged controls, and individuals with mild cognitive impairment. Furthermore, most results presented in the main paper are from female samples only unless indicated otherwise, as we observed significant sex differences in the transcriptomic data consistent with current knowledge of sex differences in LOAD[55,56]. For replication, we also consider microarray data generated from the Illumina DASL gene expression platform from the Mayo eGWAS study from TCX for N = 186 patients, of which 108 were neuropathologically confirmed AD Cases and 78 were controls[23]. Probes were

mapped to genes using BioMart. Data were adjusted for plate using ordinary least squares regression prior to manifold learning.

**Manifold learning using discriminative dimensionality reduction tree (DDRTree)**. DDRTree is a manifold learning algorithm that infers a smooth low-dimensional manifold by an approach called reverse graph embedding. Briefly, the algorithm simultaneously learns a nonlinear projection to a latent space where the points lie on a spanning tree. A reverse embedding is also simultaneously learned from the latent space to the high-dimensional data. The DDRTree algorithm can be posed as the following optimization problem:

$$\sum_{i=1}^{N} \|\mathbf{x_i} - \mathbf{W z_i}\|^2 + \frac{\lambda}{2} \sum_{k,k'} b_{k,k'} \|\mathbf{W y_k} - \mathbf{W y_{k'}}\|^2 + \gamma \left[ \sum_{k=1}^{K} \sum_{i=1}^{N} r_{i,k} \|\mathbf{z_i} - \mathbf{y_k}\|^2 + \sigma r_{i,k} \log(r_{i,k}) \right],$$

s.t. **B** represents a spanning tree,

$$\mathbf{W^T W} = \mathbf{I}, \, r_{i,k} \geq 0, \, \sum_{k=1}^{K} r_{i,k} = 1, \qquad (1)$$

here $\{\mathbf{x_i}\}_{i=1}^{N} \in R^{\text{genes}}$ represents RNA-Seq data from each patient sample, $\{\mathbf{z_i}\}_{i=1}^{N} \in R^2$ represents the latent representation of each sample as inferred by the algorithm, $\{\mathbf{y_k}\}_{k=1}^{K}$ represents the centers of clusters in the data set, $\mathbf{W} \in R^{2 \times \text{genes}}$ represents an inverse mapping from the latent space to the high-dimensional space of RNA-Seq data, $\mathbf{B} \in R^{K \times K}$ represents a spanning tree on which the centers of the clusters lie, and $\mathbf{R} \in R^{N \times K}$ captures the soft clustering information of samples in the data set. The first term of the optimization problem is responsible for learning a low-dimensional representation of the data such that an inverse mapping exists to the high-dimensional data points, the second term learns the tree structure of the points, and the third term learns a soft clustering for the latent dimension points as well as the centers of the clusters. Despite the non-convexity of the problem, each individual optimization variable can be solved for efficiently using alternative minimization[57]. This algorithm was implemented using the Monocle package in R[19]. When fitting the Monocle objects, we also considered various adjustments to the expression data prior to manifold learning for ten principal components of genetic ancestry, RIN, PMI, age, Braak score—among other potential confounds. The code to infer the lineage in Mayo RNA-seq is available here: https://github.com/Sage-Bionetworks/AMPAD_Lineage/blob/paper_rewrites_1/TCX_GenerateMonocleDS_new.R, and code used to infer the lineage in ROSMAP is available here: https://github.com/Sage-Bionetworks/AMPAD_Lineage/blob/paper_rewrites_1/DLPFC_GenerateMonocleDS_new.R. Code to perform analysis on Mayo eGWAS study is available here: https://github.com/Sage-Bionetworks/AMPAD_Lineage/blob/paper_rewrites_ben_april_2020/mayo_egwas_GenerateMonocleDS_new.R.

**Branch assignment and pseudotime calculation for samples**. Branch assignment and pseudotime calculation were also performed using the Monocle package[19]. Briefly, pseudotime is calculated by first identifying a root point on one of the two ends of the maximum diameter path in the tree. Then the pseudotime of each point is calculated by projecting it to its closest point on the spanning tree and calculating the geodesic distance to the root point. Assigning samples to branches is done by first identifying the branches of the spanning tree and then assigning samples to the branch on which their projection to the spanning tree lies on. Robustness of pseudotime was assessed with leave one out cross validation by dropping one sample at a time, running the DDRTree method with Monocle, and then computing the absolute value of the correlation between the pseudotime estimated with the reduced data set, and the pseudotime estimated with the full data set. Alternative approaches for performing dimensionality reduction included PCA, t-stochastic neighborhood embedding (tSNE)[33], and UMAP[34], which were all run on the same data set as the DDRTree method was run in R (here: https://github.com/Sage-Bionetworks/AMPAD_Lineage/blob/paper_rewrites_1/TCX_GenerateMonocleDS_new.R and https://github.com/Sage-Bionetworks/AMPAD_Lineage/blob/paper_rewrites_1/DLPFC_GenerateMonocleDS_new.R).

**Association of pseudotime with AD status, hallmarks of Alzheimer's disease, and cognitive diagnosis**. We test for association between disease pseudotime and AD case or control status with logistic regression with AD case or control status as the outcome and inferred pseudotime as the dependent variable in both the Mayo and ROS/MAP studies. We test for association between pseudotime and hallmarks of disease in the ROS/MAP studies for both Braak (measure of tau pathology) score and CERAD score (measure of amyloid pathology) with an ordinal logistic regression model, with the neuropath score as the ordered outcome, and pseudotime as the dependent variable. Finally, we test for association between disease pseudotime and cognitive diagnosis for the following ordered clinical diagnoses of no cognitive impairment, mild cognitive impairment, and probable AD with an ordinal logistic regression model. All code for running these association tests is available: https://github.com/Sage-Bionetworks/AMPAD_Lineage/blob/paper_rewrites_1/paper_figures.Rmd.

**Inferring cell-type-specific expression patterns given marker gene expression as a function of pseudotime**. List of marker genes for different major cell types in the brain was curated from a previously published brain cell expression signature study[37]. The marker gene list was then pruned to include only genes that were

included in lineage construction. Each gene's expression as a function of pseudo-time was then obtained by smoothing using a smoothing spline of degree of freedom = 3 and normalized to lie in [0,1]. The smoothing was done to remove the effects of technical noise introduced due to RNA-Seq and the normalization was done since the absolute expression levels of genes might be very different from each other. The smoothed and normalized expression of marker genes for each category was then averaged to obtain the average marker gene expression as a function of pseudotime. A linear model was used to test for association between average expression of a given cell-type expression signature and pseudotime.

**Association between GWAS loci and correlation with pseudotime.** To test for association between pseudotime and LOAD GWAS genes, we computed the Spearman's correlation between each gene's expression and pseudotime in the Mayo and ROS/MAP studies. Next, we considered the 60 highly prioritized genes (priority score > 4) identified within AD GWAS loci by the IGAP[25]. We test for a difference between the correlation with pseudotime of background of all other genes and the IGAP AD genes using a linear model and see a significant increase in correlation between gene expression and pseudotime in both the Mayo and ROS/MAP study for AD GWAS genes.

**Branch-specific differential expression analysis.** We perform a state-specific differential expression analysis using a one-way ANOVA model in both the Mayo and ROS/MAP studies. The branch with the highest proportion of AD controls is defined as the reference branch for all analyses. We use Tukey's honest significant difference method to compute P values for the test for change in expression of a given gene compared to the reference branch. Genes are grouped based on their branch and direction of change in expression for further downstream pathway enrichment analyses. Overlap between differential expressed genes was depicted using UpSet plots[58] (Supplementary Fig. 24). Code to run analyses is available here: https://github.com/Sage-Bionetworks/AMPAD_Lineage/blob/paper_rewrites_1/DLPFC_DE_Anova.R for ROS/MAP and here: https://github.com/Sage-Bionetworks/AMPAD_Lineage/blob/paper_rewrites_1/TCX_DE_Anova.R for Mayo.

**Branch-specific gene expression signatures.** Branch-specific expression signature was obtained by first calculating the average normalized expression for all genes in each state/branch. This was followed by performing a biclustering using the pheatmap package in R (https://cran.r-project.org/web/packages/pheatmap/index.html), which uses hierarchical clustering on both samples and genes. We also used the pheatmap R package to visualize the state-specific expression signatures.

**Disease resistant subgroup validation analysis.** After identifying potentially disease resistant individuals in the Mayo RNA-seq study based on the TCX brain region (individuals from Branch 5, Fig. 5a), we considered the Mayo eGWAS TCX data, and defined neuropathological controls with pseudotimes in the top quintile of all pseudotimes as disease resistant (N = 9). We then performed a differential expression analysis using linear regression to identify array probes that were differentially expressed between resistant and nonresistant individuals, of which there were more than 5000 probes that were either up or down regulated at an FDR of 0.05. Overlaps were explored between the branch-specific gene clusters from Mayo RNA-seq (Fig. 5b) and these Mayo eGWAS resistance differential expressed probes using UpSet plots[58] (Supplementary Fig. 25).

**Gene set enrichment analyses.** For each branch-specific differential expression gene set (DEGs) in both Mayo RNA-seq and ROS/MAP, we perform a gene set enrichment analysis against Gene Ontology pathways using the enrichR[42] R package. Only pathways with FDR < 0.05 are reported. The code we used to run the ROS/MAP DEG enrichments is available here: https://github.com/Sage-Bionetworks/AMPAD_Lineage/blob/paper_rewrites_1/lineage.Rmd, the code we used to run the Mayo DEG enrichments is available here: https://github.com/Sage-Bionetworks/AMPAD_Lineage/blob/paper_rewrites_1/lineageTCX.Rmd, and the code we used to run the branch-specific gene expression signature pathway enrichments is available here: https://github.com/Sage-Bionetworks/AMPAD_Lineage/blob/paper_rewrites_1/resilience.Rmd.

**Whole-genome sequencing.** Whole-genome sequencing was performed at the New York Genome Center for all individuals from the ROS/MAP and Mayo cohorts. Detailed information for both data sets can be accessed via synapse (https://doi.org/10.7303/syn2580853). Briefly, 650 ng of genomic DNA from whole blood was sheared using a Covaris LE220 sonicator. DNA fragments underwent bead-based size selection and were subsequently end-repaired, adenylated, and ligated to Illumina sequencing adapters. Libraries were sequenced on an Illumina HiSeq X sequencer using 2 × 150 bp cycles. Paired-end reads were aligned to the GRCh37 (hg19) human reference genome using the Burrows–Wheeler Aligner (BWA-MEM v0.7.8) and processed using the GATK best-practices workflow[59,60]. This included marking of duplicate reads by the use of Picard tools v1.83, local realignment around indels, and base quality score recalibration via Genome Analysis Toolkit (GATK v3.4.0). Joint variant calling

files (vcfs) for whole-genome sequencing data for the Mayo and ROS/MAP cohort were obtained through the AMP-AD knowledge portal (https://doi.org/10.7303/syn10901595).

**Single variant association with pseudotime in two independent cohorts.** Likelihood ratio tests within a linear regression framework were used to model the relationship between the quantitative expression trait pseudotime and genetic variants in 436 AD cases. Genome-wide genetic association analysis was performed for 305 female patients in the ROS/MAP cohort and 131 female patients in the Mayo cohort for which both genotyping and postmortem RNA-seq data were available. An efficient mixed model approach, implemented in the EMMAX software suite, was used to account for potential biases and cryptic relatedness among individuals[61]. Only variants with MAF > 0.05, genotyping call rates > 95%, minimum sequencing depth of 20 reads and Hardy–Weinberg equilibrium $P > 10^{-4}$ were considered for analysis. Quantile-quantile plots (Supplementary Figs. 21 and 22) for the test statistics showed no significant derivation between expected and observed P values, highlighting that there is no consistent differences across cases and controls except for the small number of significantly associated variants. Furthermore, the genomic inflation factor (lambda) was determined to be 0.99 for the Mayo and 0.98 for the ROS/MAP single variant association tests. This highlights that potential confounding factors, such as population stratification have been adequately controlled.

**Reporting summary.** Further information on research design is available in the Nature Research Reporting Summary linked to this article.

## Data availability

All source data analyzed in the study are publicly available[22–24,46]. Specifically, we use a version of the RNA-seq data from the ROS/MAP study (https://doi.org/10.7303/syn8456638.22) and RNA-seq data from the Mayo RNA-seq (https://doi.org/10.7303/syn8466816.19) run through the same bioinformatic processing pipeline[22]. The array expression data from the Mayo eGWAS study are available at: https://doi.org/10.7303/syn3617054.1. Pseudotimes for ROS/MAP individuals are available for all individuals, females, and males, respectively (https://doi.org/10.7303/syn23446661.2, https://doi.org/10.7303/syn22822695.1, https://doi.org/10.7303/syn23446654.3). Pseudotimes for Mayo RNA-seq individuals are available for all individuals, females, and males, respectively (https://doi.org/10.7303/syn23446689.1, https://doi.org/10.7303/syn22822691.1, https://doi.org/10.7303/syn23446688.1). Pseudotimes for female Mayo eGWAS individuals are available (https://doi.org/10.7303/syn22822690.1). Source data for Fig. 2c (https://doi.org/10.7303/syn23246577.1), Fig. 2d (https://doi.org/10.7303/syn23246583.1), Fig. 3b (https://doi.org/10.7303/syn23246585.1), Fig. 5b (https://doi.org/10.7303/syn22822693.1), and Supplementary Fig. 6A (https://doi.org/10.7303/syn23445580.2), Supplementary Fig. 6B (https://doi.org/10.7303/syn23445582.3), Supplementary Fig. 6C (https://doi.org/10.7303/syn23445583.2), Supplementary Fig. 6D (https://doi.org/10.7303/syn23445584.2), Supplementary Fig. 7B (https://doi.org/10.7303/syn23246588.1), Supplementary Fig. 8C (https://doi.org/10.7303/syn23446680.1), Supplementary Fig. 8D (https://doi.org/10.7303/syn23446682.3), Supplementary Fig. 9D–F (https://doi.org/10.7303/syn23448900.1), Supplementary Fig. 10C (https://doi.org/10.7303/syn23446681.1), Supplementary Fig. 10D (https://doi.org/10.7303/syn23446691.1), Supplementary Fig. 11D–F (https://doi.org/10.7303/syn23448904.1), Supplementary Fig. 12A (https://doi.org/10.7303/syn23446257.3), Supplementary Fig. 12B (https://doi.org/10.7303/syn23446326.2), Supplementary Fig. 12C (https://doi.org/10.7303/syn23446331.1), Supplementary Fig. 12D (https://doi.org/10.7303/syn23446332.1), Supplementary Fig. 13B (https://doi.org/10.7303/syn23450641.1), Supplementary Fig. 13D (https://doi.org/10.7303/syn23448918.1), Supplementary Fig. 13E (https://doi.org/10.7303/syn23452920.1), Supplementary Fig. 14B (https://doi.org/10.7303/syn23468302.1), Supplementary Fig. 17C (https://doi.org/10.7303/syn23505161.1), Supplementary Fig. 17D (https://doi.org/10.7303/syn23505539.1), Supplementary Fig. 18D–F (https://doi.org/10.7303/syn23508896.1), Supplementary Fig. 24 (https://doi.org/10.7303/syn23246590.1), Supplementary Fig. 25 (https://doi.org/10.7303/syn23246594.1), Supplementary Fig. 28 (https://doi.org/10.7303/syn23246595.1) are also available.

## Code availability

All code is publicly available (https://github.com/Sage-Bionetworks/AMPAD_Lineage). References to code to perform specific analyses are described in detail in "Methods."

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

## Acknowledgements

This work was supported by NIA grants U54AG054345 and RF1AG057443. The ROSMAP Study data were provided by the Rush Alzheimer's Disease Center, Rush University Medical Center, Chicago. Data collection was supported through funding by NIA grants P30AG10161, R01AG15819, R01AG17917, R01AG30146, R01AG36836, U01AG32984, U01AG46152, the Illinois Department of Public Health, and the Translational Genomics Research Institute. Mayo RNA-seq Study data were provided by the following sources: The Mayo Clinic Alzheimer's Disease Genetic Studies, led by N.E.T. and Dr. Steven G. Younkin, Mayo Clinic, Jacksonville, FL using samples from the Mayo Clinic Study of Aging, the Mayo Clinic Alzheimer's Disease Research Center, and the Mayo Clinic Brain Bank. Data collection was supported through funding by NIA grants P50 AG016574, R01AG032990, U01AG046139, R01AG018023, U01AG006576, U01AG006786, R01 AG025711, R01AG017216, R01AG003949, NINDS grant R01NS080820, CurePSP Foundation, and support from Mayo Foundation. Study data include samples collected through the Sun Health Research Institute Brain and Body Donation Program of Sun City, Arizona. The Brain and Body Donation Program is supported by the National Institute of Neurological Disorders and Stroke (U24 NS072026 National Brain and Tissue Resource for Parkinson's Disease and Related Disorders), the National Institute on Aging (P30 AG19610 Arizona Alzheimer's Disease Core Center), the Arizona Department of Health Services (contract 211002, Arizona Alzheimer's Research Center), the Arizona Biomedical Research Commission (contracts 4001, 0011, 05-901, and 1001 to the Arizona Parkinson's Disease Consortium), and the Michael J. Fox Foundation for Parkinson's Research. MSBB data were generated from postmortem brain tissue collected through the Mount Sinai VA Medical Center Brain Bank and were provided by Dr. Eric Schadt from Mount Sinai School of Medicine.

## Author contributions

S.M. and B.A.L. designed the study. S.M., L.H., and B.A.L. performed the analyses. S.M., L.H., C.P., S.J., G.A.G., A.K.G., S.K.S., P.L.D.J., N.E.T., G.W.C., L.M.M., and B.A.L. wrote the paper.

## Competing interests

The authors declare no competing interests.
