## [Peer Review File · Nature Communications]

Reviewers' comments:

Reviewer #1 (Remarks to the Author):

This paper proposed a novel approach to infer the Alzheimer's disease (AD) severity and disease subtypes in an unsupervised manner from post-mortem bulk RNA-seq data. Understanding the molecular heterogeneity of AD is very important. Even though the proposed approach and some of the results are promising, it would be great if authors improve the manuscript based on the following comments.

- Previous studies of AD genomics/transcriptomics and such indicate that AD is genetically heterogeneous, with risk of late-onset AD the result of the combination of many risk factors with small effects on risk. These small effects can easily be masked by structure in the data, such as population structure and batch effects, or important covariates like post-mortem interval for transcriptomics. How will the proposed method accommodate the known structure in the raw data sets? For example, recent meta-analyses of the ADGC, IGAP, and other consortia GWAS data show clear inflation due to structure or batch effects despite efforts to adjust for center effects. In the transcriptomics analyses of AMP-AD, postmortem interval and other factors have shown to cause variation in the results unrelated to the pathogenesis of AD. How do you recognize that the results are not highlighting these unrelated correlations across individuals?

- Unsupervised manifold learning is an interesting approach to identify distinct disease subtypes. However, Fig 2B looks kind of random to me. One major limitation of manifold learning is related to the smoothness assumption. Did they try to use other unsupervised learning approaches to compare the results?

- They also performed manifold learning using only genes with high variance across samples and saw a strong concordance with disease lineages inferred with differentially expressed genes (Figure S4). However, I am curious why they used the less stringent criteria ($FDR < 0.1$) to identify differential expressed genes.

- There were many previous studies to find the AD subtypes, mostly based on imaging dataset. Any comparison with that? Did they find a novel subtype that is not found from the imaging dataset?

- What is CNS transcriptomic data described in the abstract?

Reviewer #2 (Remarks to the Author):

In this manuscript the authors apply manifold learning to develop a molecular model for temporal disease progression in Alzheimer's disease based on post mortem brain RNA-Seq data. They show that the estimate of the disease progression is concordant with several tau and amyloid pathology scores as well as cognitive diagnosis. They investigate association with cell type abundance and genes that harbor known AD loci. Finally they identify a set of control samples which are estimated to have "late-stage" disease. The paper is clearly written and the figures are informative. The application of manifold learning to estimate disease progression is interesting and novel, however there are some additional analyses would strengthen the manuscript significantly.

Major Comments:

1) Traditional visualization of the samples using PCA/TSNE/UMAP based on both all genes and differentially expressed genes would be informative. Additional analysis of how the traditional approaches compare to manifold learning in terms of correlation of PCs with tau and amyloid pathology and cognitive diagnosis will demonstrate the potential utility of the manifold learning

approach.

2) I think differences between the analyses across the two brain regions should be explored further. There are four branches identified in TCX and 5 in DLPFC. What is the overlap like in the differentially expressed genes and pathways across the branches? Is there a dataset that the authors can pull in which contains samples from the two regions in the same patients? Are the temporal annotations concordant?

3) The authors mention that there is sex heterogeneity when performing the manifold learning approach, and they choose to focus on the female samples only however the differences are not explored. To validate the approach further, it would be informative to see the analysis joint on both males and females as well as a stratified analysis by sex.

4) Figure 1 would benefit from additional details on the methodology at each step. What are the parameters/methods applied – how is dimensionality reduced? Data from which brain regions is being captured? How many samples? Similar details should be added to the results section. Also the full set of analyses should be included in Figure 1 to give an overall workflow of the paper including the cell type analysis, genetics association, etc.

5) The methods section is also missing critical details, on the number of samples per study, number of genes differentially expressed / used in the manifold analysis, etc – these should be included.

6) The cell type analysis is very interesting and should be pulled into the main manuscript.

7) The APOE analysis is interesting and should be explored further. While the plots for both TCX and FLPC are shown in the supplement, the statistics are only provided for the TCX analysis.

8) Figure 3 – are the pathology scores available for TCX as well? The data should be included.

9) The outlier control group is only present in the TCX analysis – how can the authors explain this phenomenon?

10) Finally, the authors should further think of ways to validate the progression score. Are there independent datasets (microarray or RNASeq?) that they could pull into the analysis to show similar correlation with predicted disease progression?

Minor Comments:

11) Typo on line 229: "resident" should be "resistant"?

Reviewer #3 (Remarks to the Author):

This paper reports an innovative analysis of two major brain RNA-Seq datasets from ROS/MAP and Mayo Clinic to study the biology Alzheimer's disease progression. The authors apply the DDRTree algorithm and fits the gene expression profiles of these samples into a lower dimension space, with the intuition that these cross sectional observations correspond to different time points of a common progression path (with possible branches that lead to disease subtypes). The authors reported six major branches in the fitted trees, assigned a "pseudotime" score based on where the individual samples are placed on the progression tree, and showed the pseudotime score correlates with neuropathology, reduction of neurons/increase of glial cells, and genetic hits from AD GWAS. The authors also found a unique branch that has many controls, and some of the samples show distinct expression patterns that the authors explained as "resistant states".

While the idea of the analysis is novel and it is interesting to observe the various correlations reported in this study, I have some concerns regarding the study.

First, I think the authors need to provide more evidence the correlations truly correspond to disease subtypes and progression, as it could be due to sampling bias or other easier explanation. The fact that the pseudotime score correlates with cell type mixture changes and braak staging is a warning sign that most of the correlations could be explained by severity of neurodegeneration, which is a result but not necessarily equal to progression of disease. In this case the tree method simply groups

samples by the severity of their neuropathology, an interpersonal variability. This also explains the correlation with case/control status. It is hard to eliminate this possibility with the study design (unless there is truly some longitudinal data, which is hard for AD and almost impossible for brain gene expression). But the authors could check if the pseudotime still has other correlations after correcting for Braak staging and age to see if there are other novel signals, of which there are (e.g. gene expression change in IGAP genes).

There is insufficient information about the cohorts. I only found the sample count in the last part of the methods section describing genetic association. There should be more details regarding sample size, quality of RNA-Seq data, diagnosis procedure, or basic information (e.g. composition by sex, age at onset, Braak staging) about the two datasets at the beginning of the methods or results section.

How evenly collected are the samples? If they are all late stage AD (which is very likely in this study), it is hard to argue that the analysis truly captures progression of disease.

There is lack of statistics discussion regarding the tree learning method. One always gets a tree from these methods, how do we know that the learned tree structures are indeed due to data or just a bad fit, and that all 6 major branches are robust signals? Some kind of evaluation would be helpful (e.g. bootstrap).

How do the authors choose the IGAP genes? Are they just the nearest genes for the top signals (there are 20) or all genes within certain distance to the top signals? It would be worthwhile to check other nearby genes since we now know some of these genetic variations could be affecting gene up to a million basepairs away or more.

It would be helpful if the authors provide more information about the "resistant state" samples. For example did these patients truly exhibit resilience in cognitive decline?

Reviewer #1.

Previous studies of AD genomics/transcriptomics and such indicate that AD is genetically heterogeneous, with risk of late-onset AD the result of the combination of many risk factors with small effects on risk. These small effects can easily be masked by structure in the data, such as population structure and batch effects, or important covariates like postmortem interval for transcriptomics. How will the proposed method accommodate the known structure in the raw data sets? For example, recent meta-analyses of the ADGC, IGAP, and other consortia GWAS data show clear inflation due to structure or batch effects despite efforts to adjust for center effects. In the transcriptomics analyses of AMP-AD, postmortem interval and other factors have shown to cause variation in the results unrelated to the pathogenesis of AD. How do you recognize that the results are not highlighting these unrelated correlations across individuals?

We agree that the underlying structure in the raw data should be explored for potential confounding effects on our pseudotime estimates. We assessed effects of the following potential confounders on pseudotime estimates: post-mortem interval (PMI), RNA integrity number (RIN), and genetic ancestry via the first ten principal components (PCs). We included each potential confounder as a variable in the Monocle object dimension reduction step by calling the function “residualModelFormulaStr” on each factor, which is explicitly designed to “subtract the effects of ‘uninteresting’ sources of variation to reduce their impact on the clustering” (<http://cole-trapnell-lab.github.io/monocle-release/docs/>) (i.e. confounder control through regression). In no case did adjustment for the covariates in creation of the monocle object, alone or combined, markedly change the association between pseudotime and AD status and we have updated both the Manuscript text (revised manuscript lines 112-116, 128-129, 154) and included additional figures and a table to reflect this this (**Table S2, Figures S3-S4, S6, S12**). Indeed, in the case of the Mayo (TCX) data, inclusion of the covariates appeared to strengthen the association between pseudotime and AD status. In addition, the correlation between each pseudotime estimate and LOAD GWAS gene expression followed similar patterns compared to the unadjusted pseudotime-gene expression correlations (supplemental figure). Going forward, we feel confident that the unadjusted pseudotime estimate is not confounded by underlying structure in such a way that yields confounded results; if anything, the unadjusted pseudotime estimate is more likely to be slightly biased toward the null for the TCX data.

Unsupervised manifold learning is an interesting approach to identify distinct disease subtypes. However, Fig 2B looks kind of random to me. One major limitation of manifold learning is related to the smoothness assumption. Did they try to use other unsupervised learning approaches to compare the results?

In our original submission we compared the DDRTree-based method via Monocle2 (as described in the revised manuscript Lines 439-470) to both Monocle1 and another manifold learning method, diffusion pseudotime (DPT) (**Figure S1-S3** in the original submission, **Figures S26-S28** and lines 418-420 in the revised manuscript). Since then, we also assessed two other common dimensionality reduction techniques: principal component analysis (PCA) and T-

distributed Stochastic Neighbor Embedding (tSNE), along with a newer manifold learning technique, Uniform Manifold Approximation and Projection (UMAP) in a new results section in the revised manuscript (revised manuscript lines 178-193). Components from all three methods were correlated with our original measure of pseudotime to varying degrees (revised manuscript **Figures S15-S16**), while lineages derived using UMAP in the newest version of Monocle3 yielded pseudotime associations that were highly similar to our original results (revised manuscript lines 190-193, **Figures S17-S18, Table S4**). While these alternative approaches each have their strengths and all yield somewhat similar results at a coarse level, we believe lineages and pseudotime estimates derived from manifold learning techniques are more apt to allow for novel insights through observation of branching patterns (such as the “resistant” state individuals observed at the end of the TCX lineage highlighted here), that would oftentimes be obscured by other dimension reduction techniques. We also see that the manifold learning approaches (both DDRTree and UMAP) have the strongest associations with disease diagnosis and hallmarks of disease compared to PCA or tSNE (revised manuscript **Table S4**). In addition, the manifold learning approaches are more intuitive methods for assessing and ordering subtle changes across complex data types, allowing for easier communication of results and higher impact within the research community.

They also performed manifold learning using only genes with high variance across samples and saw a strong concordance with disease lineages inferred with differentially expressed genes (Figure S4). However, I am curious why they used the less stringent criteria (FDR < 0.1) to identify differential expressed genes.

We used a less conservative FDR threshold in order to maximize the number of genes included in the analysis, with a reasonable expectation that we wouldn't include a large number of false positives. With the FDR threshold at 0.1, 2820 differentially expressed genes were included in the DLPFC analysis and 7234 in TCX, and resulted in the analyses presented. When we reduced the FDR threshold to 0.01, 671 genes were included in the DLPFC data set, and 4252 genes in the TCX data set. Even with a large reduction in number of DEGs included at the stringent FDR<0.01 threshold, resulting pseudotimes were highly correlated with pseudotimes calculated with the larger set of DEGs at FDR<0.1: Spearman's rho = 0.93 for TCX pseudotime correlations and Spearman's rho = 0.91 for DLPFC pseudotime correlations (revised manuscript **Figure S2**, Lines 110-111). Given the high concordance between pseudotimes, we feel confident that resulting lineages using DEGs identified under the higher FDR threshold are robust.

There were many previous studies to find the AD subtypes, mostly based on imaging dataset. Any comparison with that? Did they find a novel subtype that is not found from the imaging dataset?

Imaging data (typically via MRI or FDG-PET) has been instrumental in refining AD phenotypes and determining mechanisms of disease (for instance, through identification and quantification of factors such as hippocampal volume, TDP-43 distribution, presence of Lewy Bodies, level of cerebrovascular disease, white matter hyperintensities, etc.). Our lineage estimations would not

be able to identify subtypes at such a granular level, though we do believe it would be very interesting to see if pseudotime was associated with any MRI-identified groups, as a more specific measure of neuropathology. However, no brain imaging data, from either study, was available to us at this time and so we were not able to directly assess whether pseudotimes or lineage-derived groups estimated from differential gene expression data had any corresponding associations with phenotypic groups identified by imaging data.

What is CNS transcriptomic data described in the abstract?

We apologize for failing to clarify the abbreviation in the final sentence of the abstract. CNS stands for central nervous system. We have restated the sentence to be more consistent with language used in the rest of the abstract, lines 45-46: "In summary, we present a disease specific method for ordering patients based on their LOAD disease progression from brain transcriptomic data."

Reviewer #2:

In this manuscript the authors apply manifold learning to develop a molecular model for temporal disease progression in Alzheimer's disease based on post mortem brain RNASeq data. They show that the estimate of the disease progression is concordant with several tau and amyloid pathology scores as well as cognitive diagnosis. They investigate association with cell type abundance and genes that harbor known AD loci. Finally they identify a set of control samples which are estimated to have "late-stage" disease. The paper is clearly written and the figures are informative. The application of manifold learning to estimate disease progression is interesting and novel, however there are some additional analyses would strengthen the manuscript significantly.

Major Comments:

1) Traditional visualization of the samples using PCA, TSNE, UMAP based on both all genes and differentially expressed genes would be informative. Additional analysis of how the traditional approaches compare to manifold learning in terms of correlation of PCs with tau and amyloid pathology and cognitive diagnosis will demonstrate the potential utility of the manifold learning approach.

As discussed in the response to Reviewer #1, we apply PCA, tSNE, and UMAP to both the Mayo RNAseq and ROS/MAP data-sets, and compare the top two recovered dimensions to the pseudotimes inferred using Monocle2 (revised manuscript **Figures S15-S16**), where we see that there is general concordance across dimensionality reduction approaches as reflected in edits to the text, lines 183-186: "We see the strongest correlations between PCA1 and UMAP2 and pseudotime in both data-sets, increasing our confidence that the overarching ordering of patients along a disease pseudotime is a robust characteristic of the disease progression as reflected in the gene expression changes as a function of disease, and not dependent on the underlying manifold learning approach." In fact, we see that that lineages derived using UMAP in the newest version of Monocle3 yielded pseudotime associations that are highly similar to our

original results (revised manuscript lines 190-193, **Figures S17-S18, Table S4**). Most importantly though, we see that the manifold learning approaches of UMAP and Monocle2 have more significant associations with hallmarks of disease compared to non-manifold based dimensionality reduction approaches such as PCA or tSNE (revised manuscript lines 188-190, **Table S4**).

2) I think differences between the analyses across the two brain regions should be explored further. There are four branches identified in TCX and 5 in DLPFC. What is the overlap like in the differentially expressed genes and pathways across the branches? Is there a dataset that the authors can pull in which contains samples from the two regions in the same patients? Are the temporal annotations concordant?

We have included an UpSet plot to explore the overlap between the differential expressed gene sets (revised manuscript **Figure 24**), which shows all of the overlaps between the branch specific gene sets for the two brain regions. In addition to summarize the similarities between the two brain regions, we have added the following text, lines 253-258: "...find significant overlaps in branches enriched for late stage disease cases, especially between up-regulated genes in State 6 of DLPFC and up regulated genes in State 5 of TCX (P-value: 4.1×10^{-108} , OR: 4.5, Fisher's exact test), as well as genes that are up-regulated in State 6 of DLPFC and in State 4 of TCX (P-value: 1.8×10^{-14} , OR: 1.9, Fisher's exact test), and more modestly for genes that are down regulated in State 6 of DLPFC and down regulated in State 3 of TCX (P-value: 1.1×10^{-6} , OR: 1.6, Fisher's exact test)." We interpret this as there being strong similarities in the late stage disease characteristics across data-sets. Unfortunately there is a lack of power to detect similarities for early stage disease branches (many early branches do not have differential expressed genes, especially in DLPFC). Unfortunately we are not aware of an available data-set that has samples from these two regions in the same patients, so were unable to perform the second suggested analysis. Finally, the robust associations between disease pseudotime and AD case/control status, Braak, CERAD, and cogdx in not only the Mayo RNAseq and ROS/MAP data (revised manuscript **Figure 2-3**), but also in the Mayo eGWAS study (revised manuscript **Figure S7**) increase our confidence in the concordance of temporal annotations of applying this approach.

3) The authors mention that there is sex heterogeneity when performing the manifold learning approach, and they choose to focus on the female samples only however the differences are not explored. To validate the approach further, it would be informative to see the analysis joint on both males and females as well as a stratified analysis by sex.

As described in the manuscript, we chose to focus on samples obtained from women for the primary analysis and presentation of results, since previous work has shown significant sex differences in transcriptomic data that is consistent with current knowledge of sex differences in LOAD, revised manuscript lines 139-140: "Furthermore, we observe strong evidence of sex heterogeneity when performing the manifold learning approach, and find that the manifolds inferred for female only samples show stronger association with pseudotime than for male samples. This matches previous observations concerning disease specific sex heterogeneity²²."

We have included the male-only and combined analyses in supplementary data for comparison (revised manuscript lines 140-143, **Figures S8-S9**). In general, pseudotimes calculated from male samples had higher variance and showed weaker associations with clinical phenotypes (revised manuscript **Figures S8-S9**), though the direction of the associations was consistent with the results observed in female-only analyses. Analyses using combined samples were also consistent with the female-only analyses (revised manuscript lines 143-147, **Figures S10-S11**).

4) Figure 1 would benefit from additional details on the methodology at each step. What are the parameters/methods applied – how is dimensionality reduced? Data from which brain regions is being captured? How many samples? Similar details should be added to the results section. Also the full set of analyses should be included in Figure 1 to give on overall workflow of the paper including the cell type analysis, genetics association, etc.

We have updated Figure 1 to indicate more methodological details of the manifold learning, approach as well as sample numbers and brain regions, and downstream analyses. (Revised manuscript, **Figure 1**). Furthermore, we have included a much expanded breakdown of the data used in the study in **Table S1**, including details on the number of samples, the demographics, the distribution of neuropathology, among other pertinent variables.

5) The methods section is also missing critical details, on the number of samples per study, number of genes differentially expressed / used in the manifold analysis, etc – these should be included.

We have included a paragraph in the methods section describing the cohorts, revised manuscript lines: 355-377 (which includes sample numbers and number of differentially expressed genes used in the manifold analysis, among other characteristics per Reviewer #3), and is summarized in **Table S1**.

6) The cell type analysis is very interesting and should be pulled into the main manuscript.
suggested edits

We have incorporated the cell type analysis in the revised manuscript, lines 195-208: “To test if the inferred pseudotime aligns with existing cell type specific hypotheses regarding AD, we first selected from the genes used in lineage construction the marker genes for four key cell types: neurons, astrocytes, microglia, and oligodendrocytes based on a previously published brain cell atlas³⁷ (**Table S5**). We then calculate the normalized mean expression for the marker genes of each cell type and fitted a linear model to the mean expression with disease pseudotime as the dependent variable. We find that, in both studies, the cell specific marker gene levels show a statistically significant linear dependence on pseudotime (**Table S6**). Fitted effects recapitulate known neuropathologic changes which occur in AD, namely: i) a reduction in the neuronal populations as AD progresses, and ii) an increase in expression associated with activation of microglia, astrocytes, and oligodendrocytes as AD progresses (**Figure 4**).” Where we have moved the figure showing the dependence between pseudotime and cell markers into the main manuscript.

7) *The APOE analysis is interesting and should be explored further. While the plots for both TCX and DLPFC are shown in the supplement, the statistics are only provided for the TCX analysis.*

In the DLPFC region we did not observe a significant APOE e4 association for any of the branches, but we did observe a significant association between disease pseudotime and APOE e4 dosage in the new Mayo eGWAS analysis, revised manuscript, lines 210-216: “Next, we test for association between assigned lineage state in ROS/MAP (DLPFC) and Mayo (TCX) and APOE e4 status (**Figure S19**). For reference, the inferred trees for TCX and DLPFC each resolve into 6 branches (**Figure 5A,S20**). Carriers of the APOE e4 allele are significantly enriched on the State 4 branch in TCX (P-value = 0.027, unadjusted), and suggestively enriched on the State 5 branch (P-value = 0.06, unadjusted), compared to the State 1 branch (logistic regression). Similarly, in the Mayo eGWAS study, when we perform an ordinal logistic regression of APOE e4 dosage and disease pseudotime we see a significant positive association as a function of pseudotime (P-value = 6.9×10^{-4} , **Figure S7C**).”

8) *Figure 3 – are the pathology scores available for TCX as well? The data should be included.*

We have included all available pathology scores for TCX in the Patient Characteristics table in Supplementary Materials, which included Braak stage and Thal amyloid stage only. Unfortunately, this data had a high degree of missingness, for both Braak stage (27% missing) and Thal Amyloid stage (58% missing), so we did not proceed with analyses investigating lineage associations with disease staging for the TCX samples as we did in the DLPFC samples. Even given that, we performed an analysis for association between pseudotime and neuropathology in TCX, as discussed on revised manuscript lines 172-176: “In the Mayo RNA-seq study we have Braak score and Thal Amyloid scores for only a subset of samples, but observe a similar pattern as in ROS/MAP (**Figure S14A-B**) for the samples that we do have data. There is a significant association between Braak score and pseudotime (P-value: 5×10^{-5}) as well as Thal amyloid (P-value: 1.7×10^{-5}) within this subset with available neuropathology data.”

9) *The outlier control group is only present in the TCX analysis – how can the authors explain this phenomenon?*

The two groups differed not only by brain region (TCX vs. DLPFC) and by cohort characteristics (Mayo RNAseq vs. ROS/MAP), but also by different methods employed within each study group to collect transcriptomic data, which resulted in different depth of sequencing and quality control metrics. While we took measures to harmonize this data as much as was possible through a consistent bioinformatic pipeline, we acknowledge that potential discrepancies between cohorts may be due to sampling issues. We are encouraged by the fact though that we are able to replicate the outlier control group signature in an independent expression array data-set (Mayo eGWAS), revised manuscript lines 283-291: “To replicate this observation, we perform a differential expression analysis on individuals in the Mayo eGWAS study where we consider

individuals that are in the top quintile of pseudotime, but are classified as neuropathological controls as resistant individuals (**Figure S7D, Methods**). To test if these individuals also have a similar resistant molecular endophenotype, we compare the overlap between various differential expressed gene sets derived from these resistant individuals and the gene sets identified in the biclustering of the Mayo RNA-seq data (**Figure S25**). We observe that there is a highly statistically significant overlap between genes that are upregulated in these Mayo eGWAS resistant individuals (P-value: 2.6×10^{-51} , OR: 2.9, Fisher's exact test), and the Cluster 4 genes that are upregulated in the branch 5 Mayo RNA-seq samples (**Figure S25**)."

10) Finally, the authors should further think of ways to validate the progression score. Are there independent datasets (microarray or RNASeq?) that they could pull into the analysis to show similar correlation with predicted disease progression?

As discussed above, we have included an entirely new and independent expression array dataset in our analyses, and strongly replicate the disease pseudotime association with Alzheimer's disease case/control status based on expression data from the TCX brain region, revised manuscript, lines 129-135: "To assess whether the association between inferred disease pseudotime is a phenomena in only the Mayo RNAseq and ROS/MAP RNA-seq data, we also apply the lineage inference approach to expression array data from the Mayo eQTL study²³ (**Methods**). These samples are derived from a completely independent set of donors than the Mayo RNA-seq study²⁴. Similarly, we restrict to only female samples, and test for an association between inferred disease pseudotime and disease status (**Figure S7A-B**). We see a significant association between disease pseudotime and neuropathological AD diagnosis ($P = 2.2 \times 10^{-8}$)."

Minor Comments:

11) Typo on line 229: "resident" should be "resistant"?

Thank you. We have corrected this typo.

Reviewer #3 (Remarks to the Author):

This paper reports an innovative analysis of two major brain RNA-Seq datasets from ROSMAP and Mayo Clinic to study the biology Alzheimer's disease progression. The authors apply the DDRTree algorithm and fits the gene expression profiles of these samples into a lower dimension space, with the intuition that these cross sectional observations correspond to different time points of a common progression path (with possible branches that lead to disease subtypes). The authors reported six major branches in the fitted trees, assigned a "pseudotime" score based on where the individual samples are placed on the progression tree, and showed the pseudotime score correlates with neuropathology, reduction of neurons, increase of glial cells, and genetic hits from AD GWAS. The authors also found a unique branch that has many controls, and some of the samples show distinct expression patterns that the authors explained as "resistant states". While the idea of the analysis is novel and it is interesting to observe the various correlations reported in this study, I have some concerns regarding the study.

First, I think the authors need to provide more evidence the correlations truly correspond to disease subtypes and progression, as it could be due to sampling bias or other easier explanation. The fact that the pseudotime score correlates with cell type mixture changes and braak staging is a warning sign that most of the correlations could be explained by severity of neurodegeneration, which is a result but not necessarily equal to progression of disease. In this case the tree method simply groups samples by the severity of their neuropathology, an interpersonal variability. This also explains the correlation with case/control status. It is hard to eliminate this possibility with the study design (unless there is truly some longitudinal data, which is hard for AD and almost impossible for brain gene expression). But the authors could check if the pseudotime still has other correlations after correcting for braak staging and age to see if there are other novel signals, of which there are (e.g. gene expression change in IGAP genes).

We acknowledge that our pseudotime estimate may be largely tracking neuropathology, especially given the strong association between pseudotime and Braak score (**Figure 3B**). Subtracting out these effects is difficult given the cross-sectional study design (as mentioned by the reviewer), and since diagnoses are partially dependent on Braak score and other measures of neurodegeneration, we would expect to see an attenuation of signal if we corrected for neuropathology. Indeed, subtracting Braak score from the lineage estimation process (i.e. adjusting for Braak score in the dimension reduction step) attenuates the association between pseudotime and case/control status significantly - revised manuscript lines: "In ROS/MAP, adjustment for Braak score when fitting the DDRTree method attenuates the association between pseudotime and disease states (P-value: 0.214, **Figure S13A-B**), though there is still evidence of association with cognitive diagnosis (P-value: 0.03, **Figure S13C-D**)." It also weakens differences in correlation between pseudotime and IGAP gene expression vs non-IGAP gene expression. However, though associations were not significant after adjusting for Braak score, the observed associations are still in the same direction as the original unadjusted associations. Pseudotime estimates adjusted for age at death were highly correlated with unadjusted Pseudotime estimates and did not appreciably change results. In addition, as part of this reviewer response, we adjusted for RIN number, a measure of RNA quality that is itself often correlated with level of neurodegeneration (and ensuing changes in cell type and quality), and we did not find RIN to be a significant confounder. Finally, we've included a discussion of the potential advantage of pseudotime, even if it cannot be truly interpreted as a direct measure of disease progression, revised manuscript lines 301-312: "While one could argue that the method is merely classifying patients as either disease cases or controls based on expression signatures of the hallmarks of disease, we see at least three advantages of this approach beyond that interpretation. First the application of this method appears to be produce a more quantitative measure of disease state than strictly neuropathological assessments - as born out through the identification of novel and distinct genetic loci that replicate based on IGAP summary statistics. This suggests that it may be adding information related to other aspects of disease such as the effect of neuroinflammation or neuronal injury. In addition we see evidence of neuropathological controls that are disease resistant given their molecular state in two independent studies – which would not be detectable with standard neuropathological or clinical assessments – and could provide important molecular clues to mechanisms of disease

resistance. Finally, there is the potential that specific pathways associated with early stage disease processes can be characterized which is desperately needed for hypothesis generation in the field.”

There is insufficient information about the cohorts. I only found the sample count in the last part of the methods section describing genetic association. There should be more details regarding sample size, quality of RNA-Seq data, diagnosis procedure, or basic information (e.g. composition by sex, age at onset, Braak staging) about the two datasets at the beginning of the methods or results section.

We have included a detailed summary of the patients included in the analysis in **Table S1**, along with descriptions of the clinical covariate indicator for each study in the Methods section (revised manuscript lines 355-377). Regrettably, since the age at first AD diagnosis was not included in the Mayo data, and was inconsistently collected with a high degree of missingness in the ROSMAP study, we did not include it either as a stand-alone variable in the analyses, nor as a marker on which to base age of onset.

How evenly collected are the samples? If they are all late stage AD (which is very likely in this study), it is hard to argue that the analysis truly captures progression of disease.

As indicated in the patient characteristics summary table, the samples cover a range of clinical stages by multiple measures. While time-series data would be more definitive, and control samples are admittedly not as well represented as disease-state samples (AD or other forms of dementia), we believe the diagnostic data provided by each study (i.e. clinical diagnoses, Braak staging, CERAD scores, etc.) does indicate a wide diversity of disease progression.

There is lack of statistics discussion regarding the tree learning method. One always gets a tree from these methods, how do we know that the learned tree structures are indeed due to data or just a bad fit, and that all 6 major branches are robust signals? Some kind of evaluation would be helpful (e.g. bootstrap).

We attempted to apply the bootstrap, but ran into issues with the robustness of the algorithmic implementation in that sampling with replacement caused the code to fail. Instead, we applied leave one out cross validation (or the Jackknife), and saw very robust absolute correlations between the pseudotimes estimated in downsampled data-sets and the full data-set, as described in the revised manuscript lines 116-119: “Furthermore, to assess the general robustness of the results we apply leave one out cross validation to infer disease pseudotime for both DLFP and TCX brain regions and find strong correlations between lineages inferred with each sample removed, and the lineage for the entire sample set (**Figure S5**).”

How do the authors choose the IGAP genes? Are they just the nearest genes for the top signals (there are 20) or all genes within certain distance to the top signals? It would be worthwhile to check other nearby genes since we now know some of these genetic variations could be affecting gene up to a million basepairs away or more.

As described in the supplementary table legends for **Table S3**, we pulled the reported genes from Tables 1-3 in the Kunkle et al. 2019 paper. To our knowledge these genes represented the highest prioritized genes within the known replicated AD GWAS loci from that study. We agree that there may be other genes that are farther away that could be relevant, but decided to use this conservative and established set of high prioritized genes for this work.

It would be helpful if the authors provide more information about the “resistant state” samples. For example did these patients truly exhibit resilience in cognitive decline?

We hypothesized that these individuals represent a disease-resistant state based on the lineage and gene expression patterns only. Pseudotime estimates were generally highly associated with AD case status, such that on average, individuals with AD had significantly higher pseudotime estimates than controls (Figure 2C). However, the individuals clustered at the left side of the tree and farthest from the root (Figure 4, branch 5), thus having the highest pseudotimes, were mostly AD-free: of the 12 individuals, three had been diagnosed with AD, 1 with PSP, 1 had pathological aging, and 7 were control patients. In addition, we did note that this branch was suggestively enriched with APOE4 carriers compared to the state 1 branch ($p=0.06$), and furthermore, this branch was directly connected to branch 4, which was predominantly made up of AD individuals. Unfortunately, the available neuropathology data for Mayo samples was too sparse to determine whether this cluster had Braak stage or Thal Amyloid score that differed from other clusters, and there was no available cognitive testing data, so we cannot make a further determination of true resistance in these individuals. Fortunately, at the very least as discussed above, we were able to replicate this pattern in the independent Mayo eGWAS TCX expression array data-set, revised manuscript lines 283-291: “To replicate this observation, we perform a differential expression analysis on individuals in the Mayo eGWAS study where we consider individuals that are in the top quintile of pseudotime, but are classified as neuropathological controls as resistant individuals (**Figure S7D, Methods**). To test if these individuals also have a similar resistant molecular endophenotype, we compare the overlap between various differentially expressed gene sets derived from these resistant individuals and the gene sets identified in the biclustering of the Mayo RNA-seq data (**Figure S25**). We observe that there is a highly statistically significant overlap between genes that are upregulated in these Mayo eGWAS resistant individuals (P-value: 2.6×10^{-51} , OR: 2.9, Fisher’s exact test), and the Cluster 4 genes that are upregulated in the branch 5 Mayo RNA-seq samples (**Figure S25**).”, suggesting that it is worth investigating in future work.

REVIEWERS' COMMENTS:

Reviewer #1 (Remarks to the Author):

The manuscript has been well-improved based on the comments. It would be of interest to many people in the field.

Reviewer #2 (Remarks to the Author):

The authors have addressed my comments and the manuscript is much improved.

Reviewer #3 (Remarks to the Author):

The authors have added many additional analyses and an independent dataset to replicate their methodology, demonstrating the robustness of their approach. I have reviewed and am satisfied with all the responses.